# Biological Effects of β-Glucans on Osteoclastogenesis

**DOI:** 10.3390/molecules26071982

**Published:** 2021-04-01

**Authors:** Wataru Ariyoshi, Shiika Hara, Ayaka Koga, Yoshie Nagai-Yoshioka, Ryota Yamasaki

**Affiliations:** 1Department of Health Promotion, Division of Infections and Molecular Biology, Kyushu Dental University, Fukuoka 803-8580, Japan; r16hara@fa.kyu-dent.ac.jp (S.H.); r20koga@fa.kyu-dent.ac.jp (A.K.); r16yoshioka@kyu-dent.ac.jp (Y.N.-Y.); r18yamasaki@kyu-dent.ac.jp (R.Y.); 2Department of Health Promotion, Division of Developmental Stomatognathic Function Science, Kyushu Dental University, Fukuoka 803-8580, Japan; 3School of Oral Health Sciences, Kyushu Dental University, Fukuoka 803-8580, Japan

**Keywords:** β-glucans, osteoclastogenesis, immunoreceptors, bone metabolism

## Abstract

Although the anti-tumor and anti-infective properties of β-glucans have been well-discussed, their role in bone metabolism has not been reviewed so far. This review discusses the biological effects of β-glucans on bone metabolisms, especially on bone-resorbing osteoclasts, which are differentiated from hematopoietic precursors. Multiple immunoreceptors that can recognize β-glucans were reported to be expressed in osteoclast precursors. Coordinated co-stimulatory signals mediated by these immunoreceptors are important for the regulation of osteoclastogenesis and bone remodeling. Curdlan from the bacterium *Alcaligenes faecalis* negatively regulates osteoclast differentiation in vitro by affecting both the osteoclast precursors and osteoclast-supporting cells. We also showed that laminarin, lichenan, and glucan from baker’s yeast, as well as β-1,3-glucan from *Euglema gracilisas,* inhibit the osteoclast formation in bone marrow cells. Consistent with these findings, systemic and local administration of β-glucan derived from *Aureobasidium pullulans* and *Saccharomyces cerevisiae* suppressed bone resorption in vivo. However, zymosan derived from *S. cerevisiae* stimulated the bone resorption activity and is widely used to induce arthritis in animal models. Additional research concerning the relationship between the molecular structure of β-glucan and its effect on osteoclastic bone resorption will be beneficial for the development of novel treatment strategies for bone-related diseases.

## 1. Introduction

The β-glucans, a group of polysaccharides consisting of β-(1,3)-linked β-d-glucopyranosyl units as the backbone and β-(1,6)-linked branching chain, exist widely in fungi, plants, some bacteria, seaweeds, and cereals [1,2]. While the anti-tumor, anti-infective, and immunomodulatory activities of β-glucans have been well discussed [3,4,5,6,7], their role in bone metabolism has not been reviewed.

Bone remodeling is essential for bone tissue homeostasis and involves the removal of the old bone followed by its subsequent replacement with the newly formed bone. Bone remodeling is strictly coordinated by the bone-forming osteoblasts [8] and bone-resorbing osteoclasts [9]. Osteocytes, which act as mechano-sensors in the bone tissue, are also responsible for bone remodeling [10]. Among these cells, osteoclasts have received significant attention as the target cells for skeletal diseases, and there is accumulating evidence that the modification of ostetoclastogenesis by several molecules may lead to the development of a novel treatment strategy. In the following sections, the concise biological effects of β-glucans on osteoclast differentiation and function are presented.

## 2. Immunoreceptor-Mediated Regulation of Ostetoclastogenesis

Osteoclasts derived from hematopoietic precursors are responsible for the bone resorption, which is essential for the bone remodeling process [11]. Receptor activator of nuclear factor kappa B ligand (RANKL), a type II membrane protein, is expressed in several cells including osteoblasts [12] and osteocytes [13]. RANKL binds to the functional receptor (RANK) on osteoclast precursors and induces osteoclast differentiation [11,14,15]. The binding of RANKL to RANK initiates the recruitment of tumor necrosis factor receptor-associated factor 6 (TRAF6), followed by activation of the canonical NF-κB pathway and mitogen-activated protein kinases (MAPK) [16,17]. Activation of these signaling pathways is crucial for induction of the nuclear factor of activated T-cells, cytoplasmic 1 (NFATc1) as well as for the c-fos and calcium signaling pathways [18]. NFATc1 is the master transcription factor for osteoclastogenesis and its auto-amplification promotes the efficient induction of a number of osteoclast-specific genes.

The immune and bone regulation system share a number of molecules. Multiple immunoreceptors in innate immune cells are important for the coordinated co-stimulatory signal that regulates osteoclastogenesis and bone remodeling [19]. The immunoreceptor tyrosine-based activation motif (ITAM) containing adaptor proteins and receptors were found in osteoclast precursors as well as myeloid cells [20,21,22]. Src is one of the important factors for osteoclast activation [23,24] and works in coordination with the ITAM pathway [25]. RANKL phosphorylates and activates Src, and activated Src kinase forms a complex with spleen tyrosine kinase (Syk), leading to the phosphorylation and subsequent activation of Syk. Syk induces calcium oscillation via the activation of phospholipase Cγ (PLCγ), which is required for the activation and induction of NFATc1 in osteoclasts [26,27,28,29]. These findings indicate that the ITAM-mediated co-stimulatory signals in the immune system are required for osteoclast differentiation induced by RANKL. Progression of osteoimmunology revealed the molecular mechanisms involved in the cross-regulation of bone metabolism and immune system [30].

Osteoclast precursors from a Syk-deficient mouse failed to differentiate normally in the presence of RANKL and macrophage colony-stimulating factor (M-CSF) [31,32]. Syk deletion in myeloid cells showed reduced susceptibility to alveolar bone loss in the mice periodontal ligature model [33]. Taken together, these results indicate that several agents attenuate RANKL-mediated osteoclast formation by downregulating Syk signaling, suggesting that Syk could be a potential target for the treatment of osteoclast-related diseases [33,34,35,36,37,38,39].

## 3. β-Glucan Receptors in Osteoclasts

Several receptors are responsible for the recognition of β-glucans (Figure 1). Dectin-1 is a type II membrane receptor containing extracellular C-type lectin domain [40] and ITAM at the intracellular tail [41]. Dectin-1 recognizes several fungal pathogens by binding to β-glucans and plays a pivotal role in the innate immune responses [42]. Flow cytometric analysis revealed that dectin-1 is predominantly expressed on the surface of myeloid cells, such as monocytes/macrophages and neutrophils, especially in the alveolar region [43]. Moreover, dectin-1 expression was revealed on the cell surface of CD11b^−^/^lo^Ly6C^hi^ populations; these osteoclast precursor cells were found to be expanded in the inflammatory arthritis model [44]. While the expression of dectin-1 was reported in osteoclast precursors, its expression was not observed in the osteoblast/stromal lineage [45]. Activation of dectin-1 induces Syk and activates the ITAM downstream signaling pathway, resulting in the stimulation of inflammatory response of the macrophages [46] and dendritic cells [47]. However, the effect of the interaction of β-glucans and dectin-1 on osteoclastogenesis is unclear.

Complement receptor 3 (CR3), also termed as Mac-1, consists of the non-covalently bound integrin-α_M_ (CD11b) and integrin-β_2_ (CD18) chain. CD11b binds and recognizes β-glucans [48], while CD18 transmits the signal of CD11b to the Syk cascade [6]. CR3 was reported to show high binding ability to β-glucans and initiate cytotoxic responses and phagocytosis of human and mouse leukocytes [49,50,51]. A recent study revealed that the *Candida albicans* killing capability of neutrophils was dependent on β-glucan recognition by CR3 followed by the activation of the CR3/Syk pathway, leading to light chain 3B-II (LC3B-II) accumulation [52]. CR3 is reported as a principal receptor required for the neutrophil extracellular trap formation induced by curdlan [53]. CR3 was expressed on the tartrate-resistant acid phosphatase (TRAP) positive mononuclear osteoclasts in the bone trabecular surface [54]. During the osteoclast differentiation of bone marrow cells, CR3 was expressed in mononuclear osteoclasts, but not in multinuclear cells, suggesting that CR3 may play an important role in the early stage of osteoclastogenesis [55]. Florescence-activated cell sorting experiments also reported that murine monocyte/macrophage cell line RAW264.7 cells with low CD11b expression impaired the osteoclast differentiation ability induced by RANKL [56]. Moreover, a recent study demonstrated that CD11b promoted RANKL-induced osteoclast differentiation by stimulating the signaling pathway mediated by Syk [57]. In contrast to these findings, decreased bone mass and increased osteoclast numbers were observed in CD11-deficient mice [58]. Furthermore, the activation of CR3 signaling by fibrinogen suppressed the RANKL-induced osteoclast differentiation via the recruitment of transcriptional repressor B-cell lymphoma 6 (Bcl6), followed by the downregulation of NFATc1 [58].

Other receptors, such as toll-like receptor 2 (TLR2) and CD5, have been reported to recognize β-glucans [59]. TLRs are the cell surface proteins that directly recognize diverse ligands via the extracellular domains, followed by the activation of cytoplasmic signaling that involves the adapter myeloid differentiation factor 88 (MyD88). Binding of curdlan to TLR2 and CR3 enhances immunoreactivity and the M1 polarization of macrophages through the signaling cascade mediated by MAPK and NF-κB [60,61]. The interaction of β-glucan from baker’s yeast with CR3 and TLR2 on the surface of RAW264.7 cells also activated inflammatory responses via the MAPK and NF-κB signaling cascade [62]. On the other hand, β-glucans derived from of *Grifola frondosa* (an edible mushroom in China and Japan) showed anti-inflammatory activity induced by lipopolysaccharide (LPS) in RAW264.7 cells via interaction with TLR2 rather than dectin-1 or CR3 [63]. TLR1-9 is expressed on osteoclast progenitors, and several ligands for TLRs have been shown to regulate osteoclastogenesis [64]. *Staphylococcus aureus* peptidoglycan and *Poryhyromonas gingivalis* (periodontopathic bacteria) directly activated RANKL-induced osteoclast differentiation via the NF-κB/NFATc1 axis mediated by TLR2 [59,65,66]. The synthetic ligand for TLR2 stimulated osteoclast formation induced by RANKL via the upregulation of lectin-like oxidized low-density lipoprotein receptor-1 (OLR1) and RANK [67]. On the other hand, lipoteichoic acid derived from *S. aureus* inhibited osteoclast differentiation of bone marrow cells derived from wild-type mice, but not from TLR2-deficient mice [68].

## 4. Biological Effect of β-Glucans on Bone

### 4.1. Inhibitory Effects of β-Glucans on Osteoclast Differentiation In Vitro

Molecular biological analyses of several β-glucans derived from *Alcaligenes faecalis* (curdlan), *Saccharomyces cerevisiae* (baker’s yeast and zymosan), *Laminaria* sp. (laminarin), *Cetraria islandica* (lichenan), *Euglena gracilis*, *Aureobasidium pullulans* (black yeast), and *Hordeum vulgare* L. (barley) were performed to elucidate the bioactivity of β-glucans on osteoclastogenesis. The inhibitory effects of β-glucans on osteoclast differentiation were studied in vitro (Table 1). 

We reported that curdlan, a linear β-1,3 glucan from the bacterium *Alcaligenes faecalis*, inhibited osteoclastic differentiation, maturation, and bone resorption of bone marrow cells and RAW264.7 cells by binding to the dectin-1 receptor expressed on osteoclast precursors, followed by the downregulation of Syk signaling [45]. The interaction of curdlan with dectin-1 also showed the inhibitory effect on osteoclast differentiation via interleukin 33 (IL-33) secretion, followed by enhancement of V-maf musculoaponeurotic fibrosarcoma oncogene homolog B (MafB) expression [69]. We also found that β-glucan from baker’s yeast suppressed osteoclast differentiation by downregulating NFATc1 activation. This inhibition of NFATc1 activation by β-glucan from baker’s yeast was dependent on the suppression of NF-κB signaling and c-fos expression, the stimulation of the negative regulator of osteoclastogenesis (interferon regulatory factor 8 (Irf-8)), and degrading the Syk protein via autophagy and the ubiquitin/proteasome system [70]. Consistent with these findings, the bone marrow cells containing zymosan particles failed to differentiate into osteoclasts [71,72]. Interestingly, osteoclasts that contained zymosan particles have a potential to form ruffled boarder and resorption pits on dentin slices, suggesting that zymosan did not affect osteoclast function [71]. Together, these β-glucans seemed to suppress osteoclastogenesis at the step of osteoclast precursor differentiation into mature osteoclasts. We demonstrated the obvious inhibitory effect of laminarin, lichenan, glucan from baker’s yeast, and β-1,3-glucan from *Euglena gracilisas,* as well as curdlan, on osteoclast differentiation from bone marrow cells. However, glucan from black yeast and β-d-glucan from barley showed a lesser inhibitory effect on osteoclast differentiation compared with other β-glucans (Figure 2). We have no explanation for these discrepancies; however, it is possible that these results reflect differences of purity and three-dimensional (3D) structure (e.g., β-(1,6)-linked side chains) of each of the β-glucans (Table 2). It is known that a certain amount of molecular weight is required for the biological activity of β-glucans. A recent study reported that a split-luciferase complementation assay is useful strategy to characterize the side chain structure of β-glucans [73]. Structural analyses of β-glucans are also currently under investigation in our laboratory.

**Table 1 molecules-26-01982-t001:** Inhibitory effects of β-glucans on osteoclast differentiation in vitro.

β-Glucan	Cell	Receptor	Effect	Molecular Mechanisms	References
Curdlan	BMCs RAW264.7	Dectin-1	Direct	Suppression of NFATc1 activation by down-regulation of Syk signaling	[45]
Curdlan	BMCs	Dectin-1	Direct	Suppression of NFATc1 activation by stimulation of MafB induced by IL-33	[69]
β-glucan from baker’s yeast	BMCs RAW264.7	Dectin-1	Direct	Suppression of NFATc1 activation by down-regulation of NF-κB and c-fos, stimulation of Irf-8, and induction of autophagy and ubiquitin/proteasome-mediated Syk protein degradation	[70]
Zymosan	BMCs	TLRs	Direct	Unknown	[71,72]
Curdlan (low MW)	BMCs cultured with osteoblasts	TLR2 TLR6	Indirect	Suppression of RANKL expression on osteoblasts	[74]

BMCs: bone marrow cells; NFATc1: nuclear factor of activated T-cells, cytoplasmic 1; Syk: spleen tyrosine kinase; MafB: V-maf musculoaponeurotic fibrosarcoma oncogene homolog B; IL-33: interleukin 33; NF-κB: nuclear factor kappa B; Irf-8: interferon regulatory factor 8; TLRs: toll-like receptors; MW: molecular weights; RANKL: receptor activator of nuclear factor kappa B ligand.

It was also shown that low-molecular-weight curdlan (MW 3000 kDa) suppressed osteoclast differentiation from mouse bone marrow cells, indirectly induced by RANKL via the TLR2/TLR6 signaling pathways in primary osteoblastic cells [74]. These studies indicated that curdlan potentially downregulates RANKL-induced osteoclastogenesis by affecting both the osteoclast precursors and osteoclast-supporting cells.

### 4.2. Inhibitory Effects of β-Glucans on Bone Resorption In Vivo

Significant evidence concerning the inhibitory effect of β-glucan on bone resorption was demonstrated in the in vivo animal models (Table 3), especially in the field of dental science. Oral administration of polycan derived from *Aureobasidium pullulans* attenuated alveolar bone loss, osteoclast numbers, and concentrations of inflammatory cytokines, such as interleukin 1β (IL-1β) and tumor necrosis factor α (TNF-α), induced by ligature placement in rats [75]. Other researchers showed that topical administration of a mixture of polycan and calcium gluconate significantly inhibited the bacterial proliferation, IL-β expression, and alveolar bone loss induced by ligature placements in rats [76]. Furthermore, in ovariectomized mice, oral administration of an extracellular polymer derived from *A. pullulans*, which contained 40% β-glucan [77] mixed with the leaf extract of *Textoria morbifera*, significantly reduced the osteoporotic symptoms [78].

As with the polycan, β-glucan derived from *Saccharomyces cerevisiae* reduced alveolar bone loss in diabetic rat models with periodontal disease via the downregulation of RANKL and upregulation of osteoprotegerin (OPG) [79,80] The Wistar rats that were administered soluble β-1,3/1,6-glucan from *S. cerevisiae* showed the suppression of periodontal bone loss induced by tooth ligature. Moreover, the plasma level of the hypothalamic-pituitary-adrenal (HPA) axis, TGF-β, and interleukin 10 (IL-10), which suppress osteoclast differentiation induced by LPS challenging, were significantly enhanced in rats treated with the soluble β-1,3/1,6-glucan [81].

### 4.3. Effects of β-Glucans on Bone Regeneration and Bone Metabolism

In addition to the protective activities for osteoclastic bone resorption, the biological effect of β-glucans on bone regeneration were also reported both in vitro and in vivo. A fabricated scaffold composed of curdlan, chitosan, and hydroxyapatite promoted adhesion, proliferation, alkaline phosphatase (ALP) activity, calcium deposition, and mineralized nodule formation in osteoblasts without affecting the proinflammatory cytokine secretion [82,83,84,85,86,87]. Consistent with these findings, the implantation of the composite composed of elastic hydroxyapatite and curdlan into the bone defect site in patients with long bone fracture assisted bone regeneration without the appearance of inflammation [88]. Furthermore, a four week administration of a mixture of polycan and calcium gluconate improved bone metabolism, as indicated by increased biochemical bone formation markers (bone-specific ALP, serum calcium, and serum phosphorus) and reduced biochemical bone resorption markers (urinary deoxypyridinoline, urinary cross-linked N-telopeptide of type-1 collagen, urinary calcium, and urinary phosphorus) [89]. These results indicated that the application of β-glucans as a biocompatible strategy might be a potential candidate in bone regeneration and formation.

### 4.4. Catabolic Effects of β-Glucans on Bone and Cartilage Tissue 

Although several studies have demonstrated the biological effects of β-glucans on bone regeneration and attenuation of inflammatory bone resorption, the degenerative activity of β-glucans has also been reported. The supernatant released from mouse peritoneal macrophages, stimulated with zymosan that was derived from *S. cerevisiae,* induced the bone resorption activity in vitro, which is mainly dependent on the effect of IL-1α [90]. Another group of researchers also reported that *C. albicans*-derived soluble β-glucan activated the inflammation and multinucleation of osteoclasts, which was mediated by the interaction with dectin-1, but not with TLR-4 [91]. 

On the basis of these findings, zymosan has been widely used to induce arthritis in animal models for many years. An intra-articular injection of zymosan stimulated acute inflammation, matrix metalloproteinase-2 (MMP-2) production, loss of proteoglycan, chondrocyte hypertrophy, bone erosion, and osteophyte formation, all regulated by complementary activity in mouse knee joints [92,93,94,95,96,97,98]. Furthermore, an intraperitoneal injection of curdlan and zymosan developed spondylarthritis features, such as synovial proliferation and bone erosion in mice [99,100], and the impairment of bone healing in rat fracture model [101].

## 5. Conclusions

Accumulating evidence suggests that β-glucans downregulate osteoclast differentiation and protect bone resorption in several animal models of osteoporosis and periodontitis. A variety of studies also demonstrated that scaffolds composed of β-glucans are effective in promoting bone regeneration and formation. This positive impact of β-glucans on bone tissue has led us to expect the possibility of β-glucans being used as an effective therapeutic agent against bone diseases in the future. 

However, contrasting effects of β-glucans isolated from different sources were observed on the bone tissue. Although several studies have reported that the immunomodulating [102], anti-tumor [103], anti-diabetes [104,105], and anti-oxidant [2] activities of β-glucans are dependent on their structure, research on the biological activity of β-glucans in bone remodeling is still at the primary stage. Further studies are needed to elucidate the receptor and specific signaling pathways activated by different structures of β-glucans. The progression and evidence in the field of osteoimmunology that highlight the close relationship between the immune system and bone metabolism will help in this elucidation [106].

The pharmaceutical application of β-glucans is also limited by its purity, toxicity, viscosity, and weak solubility [7,107]. As an acceptable level of solubility is one of the most important parameters for pharmaceutical agents, improvement of the low solubility of β-glucans is required. Several studies have demonstrated a modified procedure of β-glucans production to improve its rheological parameters [7,108]. Previous studies reported that physical modifications including ultrasonication [109,110], heat degradation [111], and gamma irradiation [112] induced polymer degradation and improved solubility of β-glucans. Moreover, chemical modifications of β-glucans, such as sulfation [113,114], phosphorylation [115,116,117] and oxidation [118,119], also increase its solubility. Further extensive research is needed to validate the therapeutic potential of β-glucans in the bone-related diseases in the medical, dental, and pharmaceutical fields.

## Figures and Tables

**Figure 1 molecules-26-01982-f001:**
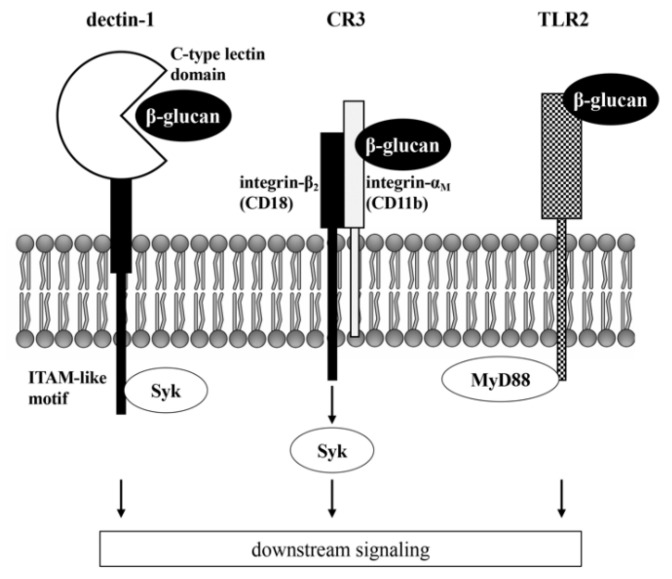
Schematic image of β-glucan recognition receptors identified in osteoclasts and their precursors.

**Figure 2 molecules-26-01982-f002:**
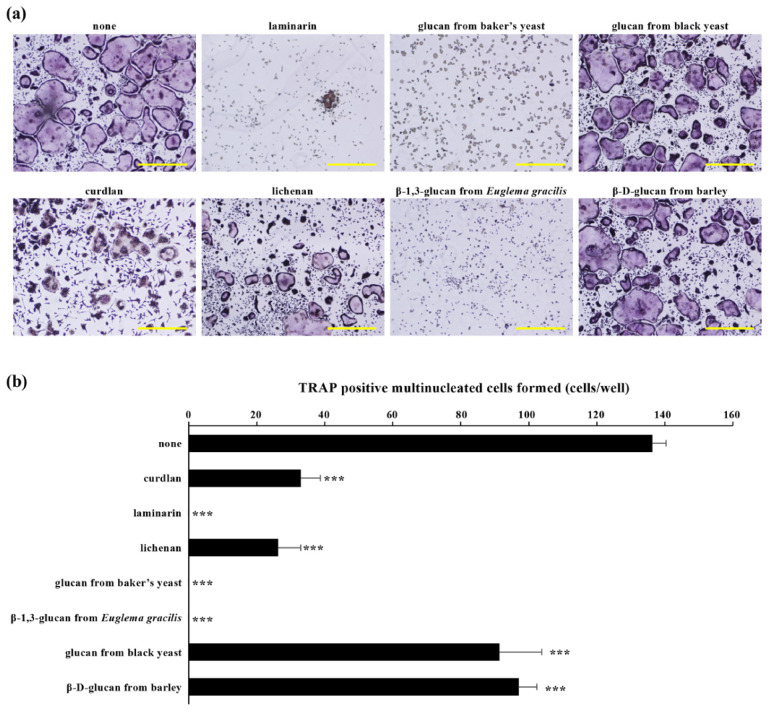
Effect of each of the β-glucans on osteoclast formation of bone marrow cells. Bone marrow cells isolated from the femurs and tibias of 6-week-old male ddY mice were incubated with macrophage colony-stimulating factor (M-CSF; 20 ng/mL) and receptor activator of nuclear factor kappa B ligand (RANKL; 40 ng/mL) in the presence or absence of each β-glucans (50 μg/mL). All the procedures were approved by the Animal Care and Use Committee of Kyushu Dental University. (**a**) Cells were cultured for four days and stained for tartrate-resistant acid phosphatase (TRAP) activity. Scale bars indicated 500 μm. (**b**) TRAP-positive multinucleated cells containing three or more nuclei were considered as osteoclasts and were counted using light microscopy. Data are presented as mean ± S.D of three independent samples. *** *p* < 0.0001 compared with the non-β-glucan treatment group (none).

**Table 2 molecules-26-01982-t002:** Source and structure of β-glucans in Figure 1.

β-Glucan	Cell	Structure
Curdlan	*Alcaligenes faecalis* var. *myxogenes*	Linear chain of β-d-(1-3)-glucopyranosyl units
Laminarin	*Laminaria* sp.	Linear chain of β-d-(1-3)-glucopyranosyl units with some 6-*O*-branching in the main chain and some β-(1,6)-intrachain links
Lichenan	*Cetraria islandica*	Linear chains of β-d-glucopyranosyl units linked via (1,3) and (1,4) linkage
Glucan from baker’s yeast	*Saccharomyces cerevisiae*	Linear chain of β-d-(1-3)-glucopyranosyl units
β-1,3-glucan from *Euglena gracilis*	*Euglena gracilis*	Linear chain of β-d-(1-3)-glucopyranosyl units
Glucan from black yeast	*Aureobasidium pullulans*	Backbone of β-d-(1-3)-glucopyranosyl units with one β-d-(1-6)-branching unit every three residues
β-d-glucan from barley	*Hordeum vulgare* L.	Linear chains of β-d-glucopyranosyl units linked via (1,3) and (1,4) linkage

**Table 3 molecules-26-01982-t003:** Inhibitory effects of β-glucans on bone loss in the in vivo animal models.

β-Glucan	Organism	Analysis	Results	References
Polycan	Male Sprague-Dawley rats	Methylene blue assay Detection of IL-1β and TNF-α Measurement of MPO activity MDA measurement iNos activity measurement Histopathology and histomorphology	Inhibited ligature-induced periodontitis and related alveolar bone loss via an antioxidant effect.	[75]
Polycan	Male SD (Crl:CD1) rats	Measurement of alveolar bone loss Microbiological analysis Measurement of MPO activity Detection of IL-1β and TNF-α MDA measurement iNos activity measurement Histopathology	Inhibited ligature-induced experimental periodontitis and related alveolar bone loss mediated by antibacterial, anti-inflammatory, and anti-oxidative activities.	[76]
Polycan	Female Sprague-Dawley rats	Detection of serum levels of osteocalcin, bALP, calcium and phosphorus Detection of urinary levels of deoxypyridinoline and creatinine Measurement of BMC, BMD and FL Histology and histomorphometry	Preserved bone mass and strength, and increased the rate of bone formation in ovariectomy-induced osteoporosis model.	[77]
β-glucan from *Aureobasidium pullulans*	Female ICR mice	Measurement of BMD, bone weight, and FL Detection of serum levels of osteocalcin and bALP Measurement of femur mineral contents Histopathology	Mixture of extracellular polymeric substances isolated from *A pullulans* and *Textoria morbifera* Nakai inhibited the ovariectomy-induced osteoporotic symptoms.	[78]
β-glucan from *Saccharomyces cerevisiae*	Male Wistar rats	Detection of β-cell function Detection of serum levels of TNF-α and IL-10 Measurement of alveolar bone loss Histometric analysis	Inhibited the systemic inflammatory profile, prevented alveolar bone loss, and improved β-cell function in streptozotocin-induced diabetic model with periodontitis.	[79]
β-glucan from *Saccharomyces cerevisiae*	Male Wistar rats	Measurement of blood glucose RT-PCR for COX-2, RANKL and OPG Morphometric analysis for alveolar bone loss	Reduced blood glucose levels and attenuated alveolar bone loss in streptozotocin-induced diabetes model with periodontitis.	[80]
Soluble β-1,3/1,6-glucan from *Saccharomyces cerevisiae*	Male Wistar rats	Radiographic examination Measurement of corticosterone Detection of serum levels of IL-10, TGF-β1 and TNF-α	Inhibited ligature-induced periodontal bone loss.	[81]

IL-1β: interleukin 1β; TNF-α: tumor necrosis factor α; MPO: myeloperoxidase; MDA: malondialdehyde; iNOS: inducible nitric oxide synthase; ALP: alkaline phosphatase; BMC: bone mineral content; BMD; bone mineral density; FL: failure load; IL-10: interleukin 10; RT-PCR: reverse transcription-polymerase chain reaction; COX-2: cyclooxygenase 2; RANKL: receptor activator of nuclear factor kappa B ligand; OPG: osteoprotegerin; TGF-β: transforming growth factor β.

## Data Availability

The data presented in this study are available on request from the corresponding author.

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
