# Peer review of "Biological Effects of β-Glucans on Osteoclastogenesis"

_molecules, 2021, doi:10.3390/molecules26071982_

Round 1
Reviewer 1 Report
This revised manuscript represents an improvement from its original version. I believe it is an interesting story and is ready for publication.
Author Response
We deeply appreciate your helpful comments.
Reviewer 2 Report
The paper reviewed biological effects of β-glucans on bone-resorbing osteoclasts. This paper discussed osteoclastic bone resorption of various β-glucans derived from Alcaligenes faecalis, baker’s yeast, Euglema gracilisas, Aureobasidium pullulans and Saccharomyces cerevisiae. The review study is interesting.
Author Response

(The authors gave the same response as above.)

Reviewer 3 Report
Manuscript ID: molecules-1141520
Type of manuscript: Review
Title: Biological Effects of β-Glucans on Osteoclastogenesis
Authors: Wataru Ariyoshi, Shiika Hara, Ayaka Koga, Yoshie Nagai-Yoshioka, Ryota Yamasaki
Submitted to section: Natural Products Chemistry,
In this review, the immunoreceptor-mediated regulation of osteoclastogenesis was explained to clarify for the reader how β-glucans can exert their effects in osteoclasts via multiple immunoreceptors. The biological effects of β-glucans on bone were presented in four parts: First, the in vitro and in vivo inhibitory effects of β-glucans on bone resorption were described. Then, the effects of β-glucans on bone regeneration, as well as on bone degeneration, were described. The authors concluded that β-glucans inhibit osteoclast differentiation and activate osteoblasts to induce bone formation. It was also mentioned in the manuscript that β-glucans can also induce degeneration in bone and cartilage tissues.
Potentially, this an important review, which may clear the complex function of β-glucans in the bone metabolismus. However, the authors should improve and modify the manuscript regarding the following statements:
1) The manuscript would be more structured, if it were organized in the following overviews:
- Introduction
- Immunoreceptor-mediated regulation of ostetoclastogenesis
- β-glucan receptors in osteoclasts
- Biological effects of β-glucans on bone
4.1. Inhibitory effects of β-glucans on osteoclast differentiation in vitro.
4.2. Inhibitory effects of β-glucans on bone resorption in vivo
4.3. Effects of β-glucans on bone regeneration and bone metabolism
4.4. Catabolic effects of β-glucans on bone and cartilage tissue
- Conclusions
2) Page 2: Please explain the abbreviations LC3B-II.
3) Immunoreceptor-mediated regulation of ostetoclastogenesis: A schematic illustration would be very helpful for the reader to understand signal transduction via immune receptor-mediated osteoclastogenesis.
4) β-glucan receptors in osteoclasts: It would also be more helpful to the reader, if a general illustration for β-glucan receptors and their downstream signaling in osteoclasts could be illustrated.
5) Page3:
- a) It is better to briefly explain all the β-glucans (curdlan, lichenan, zymosan, ...) mentioned in the manuscript with one paragraph before explaining the effect of these β-glucans on osteoclasts.
- b) Please briefly explain RAW264.7 cells.
- c) Please add the reference 58 end also the following sentence:
In contrast to these findings, decreased bone mass and increased osteoclast numbers were observed in CD11-deficient mice [58].
- d) Please explain very briefly what is "Grifola frondosa".

Author Response
Please see the attachment

This manuscript is a resubmission of an earlier submission. The following is a list of the peer review reports and author responses from that submission.
Round 1
Reviewer 1 Report
The paper reviewed biological effects of β-glucans on bone-resorbing osteoclasts differentiated from hematopoietic precursors. This paper discussed osteoclastic bone resorption of various β-glucans derived from Alcaligenes faecalis, Aureobasidium pullulans and Saccharomyces cerevisiae. In addition, this review paper discussed the relationship between the molecular structure of β-glucan and its effect on osteoclastic bone resorption. This paper cited many articles for the references. The review study is of significance, but it is much descriptive.
There is no sufficient research data presented. The authors should show some more data that can support their discussion. Figure 1 shows TRAP-positive osteoclasts in bone marrow cells.
Most parts of this paper describe bone resorption, but there is no data for bone resorption in vitro and in vivo.
The authors need to show the contents of β-glucans derived from Alcaligenes faecalis, Aureobasidium pullulans and Saccharomyces cerevisiae etc. Are the β-glucans 25 μg/ml isolated from extracts in Figure 1?
Reviewer 2 Report
Wataru Ariyoshi et al, reviewed the biological effects of β-glucans on bone metabolisms, especially on bone-resorbing osteoclasts. Although the author said the role of β-glucans in bone metabolism has not been reviewed so far, actually a review was published in 2017 (Int. J. Mol. Sci. 2017, 18(9), 1906), in which there is a whole part summarized the bone regeneration/bone Injury healing effects of β-Glucans for in vitro, in vivo and human studies. That is an integral review for the clinical and physiological functions of β-Glucans, with a lot of forms easy to read, which makes the content and novelty of this submitted review look not good enough. The good part is the author added the effect of each β-glucans on osteoclast formation of bone marrow cells by themselves, but the glucan from black yeast and from barley had almost no effect on osteoclast, and the author could not explain it. In the last section, the author mentioned the limitation of β-glucans such as the purity, toxicity, viscosity, and weak solubility, and some possible improvement, however, the author does not really summarize this part. Readers could get very few information from this short paragraph. In general, this review needs many improvements.